# Feeding Difficulties Associated with IBD during the Working Day: Qualitative Study, Alicante Spain

**DOI:** 10.3390/ijerph19063589

**Published:** 2022-03-17

**Authors:** José R. Matinez-Riera, José M. Comeche-Guijarro, Ana Gutierrez-Hervas, Sofia García-Sanjuán, Pablo Caballero

**Affiliations:** 1Department of Community Nursing, Preventive Medicine and Public Health and History of Science, University of Alicante, 03690 San Vicente del Raspeig, Spain; jr.martinez@ua.es (J.R.M.-R.); josemiguelcomeche@gmail.com (J.M.C.-G.); pablo.caballero@ua.es (P.C.); 2Department of Nursing, University of Alicante, 03690 San Vicente del Raspeig, Spain; ana.gutierrez@ua.es

**Keywords:** inflammatory bowel diseases, occupational health, diet, food, nutrition, qualitative research

## Abstract

Inflammatory bowel disease (IBD) is a chronic disease characterized by inflammation of the gastrointestinal tract. Nutrition is a key aspect as it can modulate much of the symptomatology. People affected by IBD often experience difficulties at work in all areas, including adapting their dietary management to workplace situations. The aim of this study is to identify the difficulties associated with eating during the working day in people diagnosed with IBD. A qualitative study was conducted through a nominal and focus group with people affected by IBD. After the thematic analysis of the data, three main themes emerged: management of food during the working day, searching for strategies to live with the disease, and the importance of visibility and support. This study shows that IBD influences the working life of people with IBD and makes it difficult to maintain an adequate diet. The institutions in charge of the treatment of IBD patients should consider the support of multidisciplinary teams, including nutrition professionals, as a fundamental part of the pathology control and dietary treatment to minimize its repercussions at work.

## 1. Introduction

Inflammatory bowel disease (IBD) encompasses a group of nonspecific chronic inflammatory diseases whose etiology and pathogenesis are still uncertain, including Crohn’s disease (CD) and ulcerative colitis (UC) [1]. IBD has two phases; the remission phase, where the disease usually produces no symptoms, and the flare-up phase, where IBD is active and produces various intestinal and extraintestinal symptoms, such as diarrhea, bloody stools, abdominal pain, fatigue, weight loss, or skin changes, among others [2].

The treatment of IBD is not clearly defined. It is usually treated with expensive medical treatment to reduce symptoms, which leads to a decrease in the patient’s quality of life that can affect the degree of disability and work productivity, as well as their social and personal life, producing an increase in anxiety and depression [3,4]. Nutrition is a key factor in the treatment of these patients, since nutrition can help to modulate a large part of the symptoms leading to an improvement in both disease activity and quality of life. The most characteristic symptoms are chronic diarrhea, abdominal pain, fever, rectal bleeding, alternating abdominal pain, fever, rectal bleeding, alternation of normal and diarrheal bowel movements, loss of weight and appetite, vomiting, and nausea, with periods of remission and relapses, and in the case of CD multiple hospitalizations and possible surgical intervention [5]. In addition, due to the symptomatology of the disease, people with IBD suffer feeding difficulties. They often have problems with managing their diet for fear of eating foods that will provoke a new outbreak of their disease. Therefore, malnutrition and specific nutritional deficiencies are common among this population due to several causes including malabsorption of nutrients at the intestinal level when the disease is active, hypercatabolism due to inflammatory activity, fever, corticosteroids, and infection, and decreased intake for abdominal pain, loss of hunger, malaise, and therapeutic fasting [6,7,8]. The latest evidence tries to dismantle old myths about nutrition in these patients. However, there is still insufficient evidence on the most appropriate nutrition for people with IBD [9].

With regard to occupational health, people affected with IBD often experience difficulties at work such as concentration problems, slow work pace, delays in delivery, or difficulties in working with other employees. Consequently, they may suffer from isolation at work and difficulties in decision making [3]. Similarly, flares of the disease cause these people to have higher rates of absenteeism [10] and their productivity is significantly affected, resulting in large losses in productivity costs for companies [11]. In addition, patients with more severe disease have a higher rate of unemployment or part-time work. [12]. Nevertheless, if the working environment provides facilities for IBD patients in remission or mild flares, these described problems almost disappear and they can lead a normal working life. In any case, more in-depth analyses are needed on all components related to the loss of labor productivity of these patients [13].

Therefore, providing support in the workplace by encouraging a communicative environment with other staff, as well as the opportunity for appropriate nutrition, health and wellness courses, and support groups, could have a positive impact on these patients [6,14]. Therefore, the aim of this study is to establish how eating during the working day affects people diagnosed with IBD, and the barriers and feeding strategies they have to employ to reconcile work demands.

## 2. Materials and Methods

Qualitative methodology allows us to know, describe, and understand the reality expressed by the people involved in the study. Qualitative methodology does not have external validity but it does have a strong internal validity. This means that it cannot be extrapolated to the population but it does describe in detail what happens to the study group. As a consequence, it is a methodology that emphasizes subjectivity and is interested in analyzing lived experiences and how they are interpreted by individuals. Phenomena are understood from the individual perspective. Therefore, the strength of qualitative methodology is that the researcher has access to peoples’ reasons, meanings, actions, and behaviors adopted in different scenarios and everyday situations [15,16]. Moreover, if the study group is heterogeneous, qualitative methodology will accurately capture the results of different experiences and points of view of the object of study. The main field of application of these qualitative techniques is the examination of social phenomena in their natural environment [17,18]. Therefore, qualitative methodology is the methodology for the purpose of this study, as it best illustrates the process of understanding [19].

There is not a standard approach to qualitative research based on a constructivist approach [20]. The techniques chosen for this study were Nominal Group (NG) [21,22] and Focus Groups (FG) [17,23], which are part of qualitative research.

NG and FG are forms of group interview that capitalize on communication between research participants in order to generate data [22,24]. NG and FG are collectivist, rather than individualistic, research methods and focus on the plurality and variety of participants’ attitudes, experiences, and beliefs. It is conducted in a relatively short time period [23]. While the NG technique focuses on identifying and ranking people’s needs, the FG technique identifies personal experiences on the issue.

### 2.1. Participants

The inclusion criteria for this study were: being over 18 years of age, having a definite diagnosis of IBD for at least 5 years, and being or having been employed. Due to the difficulty of finding IBD patients outside the hospital setting, the selection technique was by snowballing [25]. The research group contacted the president of the Crohn’s Disease and Ulcerative Colitis Patients’ Association of Alicante (APECCU), offered her participation in the study, and asked if she knew of any other people who might be interested with the characteristics described. Subsequently, JM C-G contacted each of the interested persons and explained in detail the characteristics of the qualitative study, introduced the team members and their experience in the qualitative technique, explained that the study was the culmination of his doctoral thesis work, and invited them to participate. Following the affirmative response of the participants, a group meeting was arranged.

Finally, 7 people participated in the study, of which 3 were women and 4 were men, aged between 26 and 53 years. All had been diagnosed with IBD for more than 5 years. In addition, 6 out of 7 of the participants were employed or on sick leave due to the COVID-19 pandemic, and one of them was also studying (Table 1). Only 4 of the 7 had a recognized disability and of these only 3 were over 33%. Participants could join either one or both meetings for the development of the NG and FG techniques.

### 2.2. Procedure

Two sessions were scheduled, the first with the NG technique and the second with the FG. In order to find out about the experience of people with IBD and how diet influences their working life, different tools were used. Firstly, a Nominal Group was used as a consensus method among the participants [26]. This process allows for the expression of personal opinions as well as interaction and discussion among the group members: it allows consensus to be reached without forcing it and identifies common areas relevant to the topic of study. After the analysis and classification of the issues or difficulties encountered, they will be compared with the available literature and the questions that will guide the second part of the method will be elaborated. On this occasion, the focus group tool will be used, with the aim of giving collective meaning to the difficulties identified in the first part of the study. There was a 20 day gap between sessions. Before the start of each session, participants were informed of the voluntary nature of the meeting and their right to leave the meeting at any time, and the anonymity of the participants was guaranteed. At this point they were given the informed consent form to be signed. The two sessions were led by professor PhD J.R. M-R. who has extensive experience in qualitative studies in both NG and FG techniques. He was assisted by PhD student and MNutr J.M.C.-G. and professor PhD PC, both male and trained by J.R.M.-R. At all times we were sensitive to issues that may affect the participants, we were careful to maintain interpersonal relationships during the conversation, and we aimed to make the meetings a positive experience for the participants as they discovered new perspectives about their own life situations [23]. These techniques ensure balanced participation of all members of the group so that we can obtain the maximum knowledge and experience from each of the participants [27,28].

Due to the confinement caused by the COVID-19 pandemic, the meetings were online. The online meetings improved the qualitative techniques in three fundamental aspects: they facilitated attendance (everyone was at home), they made it easier to exercise the right to leave the meeting (no dropouts), and they allowed recording for later verbatim transcription.

#### 2.2.1. Nominal Group Technique

In this technique, participants were invited to talk about the question “What are the difficulties in the work environment related to IBD?” They were offered the opportunity and encouraged to participate. After the discussion, assistants J.M.C.-G. and P.C. showed the group all the collected topics. Participants individually ranked the issues from most to least in terms of priority. This technique allowed multiple opinions to be obtained from different people on the topic of study, obtaining information in a structured way. In this way, a consensus method was obtained that helped in the selection of the highest priorities. Only participants and researchers were present at this meeting.

#### 2.2.2. Focus Groups Technique

Based on the results obtained in the NG and a rapid review of the literature on workers with an IBD in the last 10 years on the MEDLINE database, the research team prepared the FG meeting. The difficulties identified by the participants in the nominal group were cross-checked with retrieved literature. The preparation consisted of agreeing on questions about the target topic of this study. The agreed questions are listed in Table 2. These questions were the focus of the second meeting. In this FG technique, participants were invited to talk about the elaborated questions. The session was recorded and transcribed verbatim and turns of speech were recorded. Although no software was used to analyze the content of the session, a specific commercial software was used to transcribe the recording. Only participants and researchers were present at this meeting.

### 2.3. Data Analysis

In qualitative research, validity depends on the systematic process of obtaining and analyzing data [19]. Data were analyzed using thematic analysis. Thematic analysis is the search for repeated patterns in a verbatim transcription [29]. These patterns are encoded with small phrases or words called codes. Individually, 3 of the researchers identified on the verbatim transcription different codes in relation to the questions proposed. After this identification, the research team meets and each researcher presents their findings. Subsequently, the team discusses and triangulates the information through the different codes obtained to reach a consensus [30]. Finally, the consensus reached is compared with the available literature and theoretical frameworks.

### 2.4. Ethical Considerations

This study received a favorable evaluation from the Ethics Committee of the University of Alicante, with file number UA-2021-03-08, which was carried out in accordance with the criteria established by the Declaration of Helsinki and the WHO Good Clinical Practice Guidelines. Participation in the study was voluntary, and participants were informed that they could withdraw at any time. In order to safeguard the rights of the participants and to respect the rights of confidentiality and anonymity, pseudonyms were used in the process of recording the information for those data that allowed their identification.

## 3. Results

### 3.1. Nominal Group Results

The nominal group was composed of 6 participants. After asking the open question “What are the difficulties in the work environment related to IBD?”, the participants started to expose these difficulties. After two hours and ten minutes the information provided by the group was repeated reaching saturation. Once heard and collected, 12 difficulties were classified and related to IBD in the work environment. The results after the meeting are shown in Table 3 in order of priority.

### 3.2. Focus Group Results

The group interview saturated the information in 3 h and 20 min. Three main themes were identified from the thematic analysis of the results: (1) management of food during the working day; (2) searching for strategies to live with the disease; and (3) the importance of visibility and support (Figure 1).

#### 3.2.1. Management of Food during the Working Day

The interviewees stated that it was very difficult to adapt their meals to their workplace as, depending on what they ate, they were aware that they would have gastrointestinal disorders that would force them to go to the bathroom more frequently than their colleagues and would make it difficult to complete their tasks. This situation, together with the sick leave or the days off that they had to request for medical tests, made them feel inferior to their colleagues in opting for improvements in their jobs.

P01 *“Sometimes you don’t know if it’s agreeable to you (may upset your stomach), then of course, you may eat something that would provoke a stomachache, then not perform well at work and that affects your results.”*

P06 *“Some days you have to go to the toilet several times, if you used to go five times, now it’s 10, especially when you’re not in remission, and that is something you don’t easily get used to, because your colleagues don’t understand, and you don’t know what exactly to eat to make sure that it will not happen again.”*

P03 *“It is sometimes difficult at work, because you have to take medical tests even when you are in remission and ask for a sick leave.”*

Most of the participants mentioned the short time they had to take a break and have something to eat, either in the morning, at noon, or in the afternoon. This limitation in the schedule of meals makes it difficult to adapt their diet to their needs, which has a negative impact on adequate nutrition to maintain the disease in remission. In addition, they stated that not knowing what type of food to eat is a limitation when they have to choose what to eat outside home.

P02 *“I have Chron’s disease and, because of that, I have to eat slowly and wait a bit for digestion and at work, since I don’t have much time to eat, I just carry very little food.”*

P05 *“At lunchtime, I need as much possible time to eat, I mean, I like to have a quiet meal at lunchtime, so, if they don’t let me, I have to hurry. I prefer not to eat rather than eating in a hurry and maybe feeling sick later.”*

P04 *“It doesn’t matter what I eat because some days I feel good and some others don’t, so I try to carry small amounts and try different foods.”*

#### 3.2.2. Searching for Strategies to Live with the Disease

The interviewees express a lack of nutritional knowledge to keep their disease under control. Therefore, they rely on their beliefs, based on trial and error experiences, to modify their diet and adapt it to the state of their pathology. This learning leads them to acquire dietary habits that sometimes are not adequate for the control of their symptoms and directly affect their work development.

P05 *“I think it is a very great experience process and that, at the beginning, you are super disoriented, you don’t know where this goes, what to do, and you don’t know how to control it. You try to change your habits to try to control it and finally I think we are doing trial and error. And from there I started to make my own diet based on the symptoms that I, that I had.”*

P07 *“Because I didn’t want to eat. Because I had stomachaches and everything I ate didn’t agree with me, then I rejected food and, as nobody cared, finally I stopped eating and of course I was more tired and was unable to work.”*

Respondents who are able to plan their work schedule do so based on meals and accessibility to toilets.

P03 *“If I had an appointment with someone or a meeting with someone. Then I needed to see what the services were, go to places with enough time, stress-free and try to make that everything was easy, smooth and that I would not generate any anxiety in me and I could have a quiet meal, so I always plan it before or after.”*

The interviewees mentioned the difficulty of maintaining social relations with coworkers outside of work due to the fact that on several occasions social contact takes place in restaurants or in bars over food and drinks.

P01 *“Of course, if you go out for dinner or so, well; I always avoid it because, of course, I can’t eat all the things they serve, it is that you feel ashamed because people don’t know what it is.”*

P05 *“Of course, since I can’t have coffee, or beer, or any alcohol, when your work colleagues meet to have something, I feel bad because there are always people that don’t understand your illness and make comments.”*

#### 3.2.3. The Importance of Visibility and Support

When talking to their coworkers or employers, the interviewees recognize that in society there is a lack of visibility of IBD, its symptoms, and its consequences. This lack of knowledge of IBD on the part of employers means that they do not recognize IBD as a disabling disease and therefore do not perceive the associated tax advantages. This fact increases the degree of difficulty in finding a job.

P06 *“In respect of the need to normalize Chron’s disease, I try to do my bit, then in every job I go, if I am late I tell them, I’m so sorry, I’ve gone to the toilet five times, I am late because of that, I don’t know, you just try to make it sound normal if they ask, but they don’t seem to understand.”*

P02 *“In job interviews you say, hey I’ve got this, and then you have to explain yourself what it is because they don’t know, and then they back down and do not hire you.”*

P01 *“Sometimes you have to give many explanations to your colleagues because they don’t know what it is, some other times they don’t do it because they feel ashamed and then you start losing contact.”*

Those affected by IBD also recognize the lack of visibility in the health care system and feel ignored. In fact, the interviewees recognize that they do not have a multidisciplinary team in the health system to support them in their process and highlight the lack of knowledge of the health care teams on nutritional issues.

The importance of an expert nutritionist in IBD is a constant claim by this group. They are convinced that it could help to improve their quality of life and have a positive influence at work.

P04 *“The best thing would be to have a nutritionist that would give us some advice on what to eat or not because, for example, I have a lack of iron and so, and of course there are times when I feel tired and do not perform the same.”*

P05 *“In hospital there is not a team to guide you, then you have to look for the tools to know what to eat or not, and that’s hard, above all, when you have to reconcile meals and work, and you don’t know if it is going to upset your stomach or not.”*

## 4. Discussion

The aim of this study was to discover how eating during the working day affects people diagnosed with IBD, and the barriers and food intake strategies they adopt to reconcile work demands.

Our results show how workers affected by IBD could perceive the difficulties in managing food during their working day and how it affects them. It could be that most working people with IBD are forced to develop self-taught strategies, such as trial-and-error food management. In addition, they might experience work difficulties, including sick leave, inability to reconcile work with medical visits, restricted meal times, and living with physical problems associated with the disease. This situation may develop in people with IBD an uncomfortable perception of high rates of sick leave and a higher risk of dropping out of the labor market compared with the general population.

To contextualize these findings, the situation of people with IBD in Spain must be known. The Spanish government, through the National Institute of Social Security, assigns the degree of disability to people with different pathologies by assigning a percentage. This percentage is calculated on the basis of physical, mental, or intellectual disability, in addition to complementary social factors related to their family environment and work, educational, and cultural situation that hinder their social integration. This percentage is based on the scales described in Annex I of Royal Decree 1971/1999, of December 23, for the recognition, declaration, and qualification of the degree of disability [31]. The recognition of 33% or more provides tax and labor benefits. From a labor point of view, the main aids are the adaptation of the workplace, adapting the work environment to their characteristics, taking into account their skills, competencies, possible limitations, and flexible working hours for medical visits, even in public companies, and reduction of the working day, always depending on the degree of disability accredited [32]. However, in Spain, having IBD does not imply immediate recognition of a degree of disability higher than 33%. This will depend on the degree of the disease, the sequelae, and associated comorbidities developed.

As for the Spanish health care system, the Ministry of Health published in 2021 the update on the strategy for the management of chronic diseases [33]. This strategy compiles care measures for the chronic diseases with the highest burden of disease, however, IBD does not appear among them. As a consequence, the treatment received by people affected by IBD is based on the control of severe symptomatology and not on the follow-up and prevention and attenuation of flares [34].

Thus, as in other studies, people with IBD have reported physical, emotional, and social problems such as fatigue, pain, diarrhea, decreased cognitive functioning, embarrassment, and anxiety or depression that can result in limitations that negatively impact their work productivity [35]. In addition, they have had work difficulties such as sick leave, work flexibility for medical visits, restricted schedules for meals, and associated physical problems [3,36]. This situation exacerbates their symptomatology leading to higher rates of work disability and a higher risk of dropping out of the labor market than the general population [36,37,38].

Like other chronic diseases, such as arthritis or diabetes, IBD has high rates of absenteeism from work [39]. This fact affects the expectations of job improvements and the possibility of promotions [40], as shown in the results section. However, it is known that this situation of inequality with the healthy population can be minimized or disappear when the disease is in remission or when there are facilitators in the workplace [41]. In the context described above, these facilities are not recognized until comorbidities appear, which implies that the patient has to worsen in order to have access to the aids mentioned in the law. Therefore, with this health policy, labor equality between people with IBD and the general population may be not real. Failure to achieve work goals can lead to frustration, which in people suffering from IBD translates into a possible worsening of symptoms and therefore a greater likelihood of dropping out of the labor market.

Our study shows how people affected by IBD are forced to develop self-taught strategies, such as dietary control, in order to manage their workday [42]. People affected by IBD show dietary beliefs related to the role of diet in their time management, based on trial and error, often without professional supervision, which leads them to restrict their meal times and particular foods [43,44,45]. However, these dietary practices and self-imposed restrictions, together with the lifestyle and characteristics of the working day, can contribute to malnutrition in people with this pathology, leading to a decrease in worker productivity [46,47]. To mitigate such unfounded nutritional beliefs and their pernicious restrictive behaviors, people with IBD should undergo follow-up to improve their nutritional conditions [48,49], which, as our results describe, are not accessible in the current health care system [50]. Therefore, we believe that the institutions in charge of the treatment of IBD patients should consider the need for multidisciplinary teams that include a nutrition professional as an important part in the control of the pathology and the dietary treatment together with the medical treatment [51]. The incorporation of the professional nutritionist would modify dietary habits to benefit the course of the disease and achieve a healthy working life.

This study is not without limitations; the data presented only reflect the experience of a small group of people with IBD in the province of Alicante (Spain); the data cannot be extrapolated to other places that do not share similar characteristics. This limitation is inherent to qualitative methodology, but its strength is that all aspects related to the context of the sample are contemplated, since the sample is heterogeneous and represents the profile of the majority of people affected with IBD. Due to the restrictions imposed by the COVID-19 pandemic, and the fact that IBD patients are immunosuppressed, it was not possible to form more groups to participate in the focus and nominal group techniques. These second groups are essential to determine the saturation of information, so there is a possibility that some topics may not be mentioned [52]. However, our results show the difficulties faced by people with IBD in reconciling their work with the nutritional constraints inherent to their condition. Therefore, it would be interesting to carry out a quantitative study to corroborate the hypotheses raised in this work as a future line of study.

## 5. Conclusions

This study shows that IBD influences people’s working lives and makes it difficult to maintain an adequate diet. The lack of professional support and nutritional knowledge, together with restrictions in work schedules, leads to nutritional deficits that have a direct impact on the health of those affected. The health policies developed so far in Spain are not sufficient to maintain an adequate diet in the workplace. Institutions responsible for the management of IBD patients should consider the support of multidisciplinary teams, including nutrition professionals, as an essential part of disease management and dietary management to minimize the impact on work.

## Figures and Tables

**Figure 1 ijerph-19-03589-f001:**
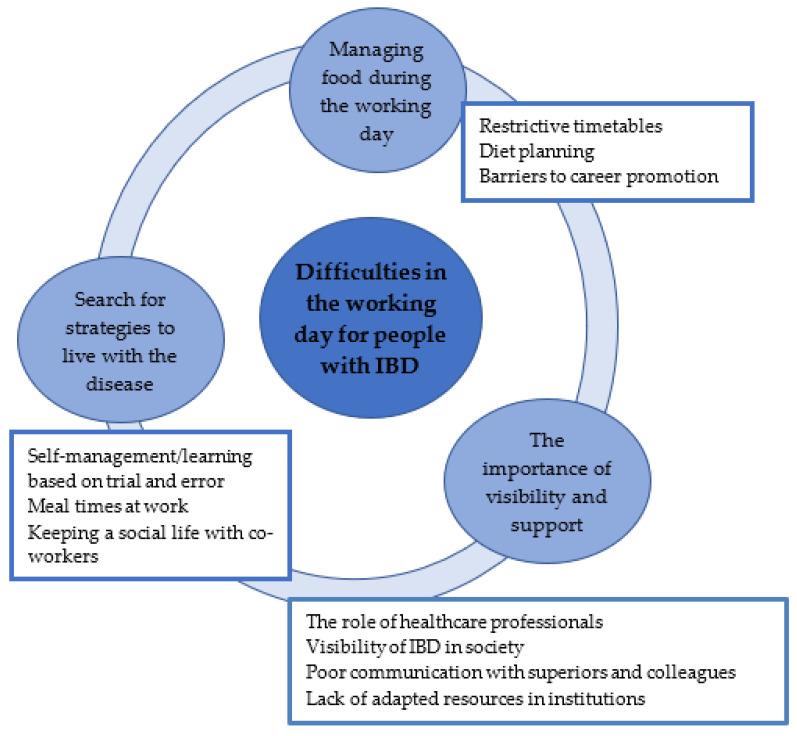
Final thematic map of themes and subthemes.

**Table 1 ijerph-19-03589-t001:** Details of participants.

Id.	Sex	Age (y.)	Type of Disease	Disease Activity	Education	Percentage of Disability	Occupation	NG	FG
1	M	28	UC	R	CH	≥33%	Junior researcher	Yes	Yes
2	M	53	UC	R	HL	≥33%	Salesman	Yes	Yes
3	M	26	CD	A	CH	<33%	School monitor	Yes	Yes
4	F	34	CD	R	HL	<33%	Unemployed	Yes	Yes
5	F	44	CD	R	HL	≥33%	Bus driver	Yes	No
6	F	41	CD	R	HL	<33%	Laboratory technician	Yes	Yes
7	M	45	UC	R	HL	<33%	Repair technician	No	Yes

F: Female; M: Male; UC: Ulcerative colitis; CD: Crohn’s disease; R: Remission; A: Active; CH: College and higher; HL: High school or less; NG: Participant of the Nominal Group; FG: Participant of the Focus Groups.

**Table 2 ijerph-19-03589-t002:** The agreed questions for Focus Group meeting.

How do you think diet can influence your work life?
How do you adapt your IBD limitations to the work environment?
Have you changed the way you eat at work because of your condition?
Have you been helped to do so?
Have you changed the way you relate to your coworkers and bosses since you had IBD?
In the institutions where you work, are they sensitive to your pathology? Is there any adaptation to your pathology at your workplace?
After being diagnosed with IBD, have you received any information or nutritional recommendations? From whom?
After being diagnosed with IBD, have you received any nutritional follow-up? Have you been diagnosed with any nutritional deficiency?

**Table 3 ijerph-19-03589-t003:** Difficulties in the work environment related to IBD.

Difficulties	Score
No fixed meal times	54
Interruptions in meal times	50
Access to personalized meals not guaranteed	49
Little time for meals	48
No access to a varied diet	45
Lack of facilities with resources for meal preparation	37
Lack of privacy and adequate access to toilets	37
No time for physical activity or rest after eating	34
No financial support for food in the workplace	31
Difficulty in adapting cooking methods to the workplace	30
No freshly cooked food	29
Lack of knowledge on the part of the company about the nutritional history of workers: allergies, intolerances, special needs, etc.	24

## Data Availability

Anonymized transcripts of interviews are available on request from the corresponding author for research purposes only.

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
