# Peer review of "Feeding Difficulties Associated with IBD during the Working Day: Qualitative Study, Alicante Spain"

_ijerph, 2022, doi:10.3390/ijerph19063589_

Round 1

Reviewer 1 Report

The authors present an interesting manuscript on perceived nutritional difficulties of IBD patients at work. As this was a qualitative study, the sample size is small, and generalizations can thus not be made – the authors fully acknowledge this in multiple parts of their manuscript. Nevertheless, the results of this study can act as important catalyst for promoting a possible policy change in how to manage IBD in Spain (e.g. by including nutritional counselling sessions for IBD patients as part of the standard of care, and by raising awareness to problems arising for such patients at work). The study could thus also lay important groundwork for a larger quantitative follow-up study, which could provide the necessary numerical data for policy makers.

I had reviewed the manuscript already when it was submitted to another journal . Back then, the authors had satisfactorily answered all my queries. They also have incorporated all my suggestions in their current version of their manuscript. In  my opinion, this manuscript is valuable and interesting, and I have no further comments. I thus recommend it to be accepted for publication subject to some minor spell and grammar checking during the editing process.

Author Response

Dear Reviewer, thank you very much for your previous contributions and support.

We will do a minor spell and grammar checking before the editing process.

Kind Regards

Reviewer 2 Report

Great and interesting work. Thank you.

Just a little comment; the use of a single nominal group perhaps subtracts some diversity from the results obtained from it. We understand that this situation derives from the fact that only 7 people participated in the study.

This numerical and territorial (Alicante) limitation should perhaps be reflected in the title of the article, since the data, as the authors say, cannot be extrapolated. Thanks

Author Response

Reviewer 2

Great and interesting work. Thank you.

  1. Just a little comment; the use of a single nominal group perhaps subtracts some diversity from the results obtained from it. We understand that this situation derives from the fact that only 7 people participated in the study.

Dear Reviewer

Thank you very much for your contributions and support

We have included in the limitations section the fact of using only one group and how it affects the results of the study.

  1. This numerical and territorial (Alicante) limitation should perhaps be reflected in the title of the article, since the data, as the authors say, cannot be extrapolated. Thanks

We have reflected this fact in the title by adding the town of Alicante to the title as you suggest.

Kind Regards

Reviewer 3 Report

  1. In the introduction, make clear the main feeding difficulties in the light of other studies;
  2. It highlights in the introduction the knowledge gap about its object of study;
  3. Nutrition is a key factor in the treatment of these patients, since nutrition can help to modulate a large part of the symptoms, leading to an improvement in both disease activity and quality of life. In addition, because of the symptomatology of the disease, malnutrition and specific nutritional deficiencies are common among people with IBD. [5,6]. Explain what "symptomatology of the disease" would be, I can't understand.
  4. About the method, explain how the studied sample took place. Was it saturation? Explain.
  5. On lines 180 to 185 explain: "colleagues and would make it difficult to fulfill their tasks"
  6. I ask the authors, why didn't they follow COREQ? 

Author Response

Reviewer 3

1. In the introduction, make clear the main feeding difficulties in the light of other studies;

 Dear Reviewer

Thank you very much for your contributions and support

 According to the reviewer's recommendations, the main feeding difficulties of IBD has been included with the corresponding reference.

2. It highlights in the introduction the knowledge gap about its object of study;

 Following the reviewer's recommendations, the lack of evidence on this topic have pointed out.

3. Nutrition is a key factor in the treatment of these patients, since nutrition can help to modulate a large part of the symptoms, leading to an improvement in both disease activity and quality of life. In addition, because of the symptomatology of the disease, malnutrition and specific nutritional deficiencies are common among people with IBD. [5,6]. Explain what "symptomatology of the disease" would be, I can't understand.

 According to the reviewer's recommendations, the main symptomatology of these pathologies has been included with the corresponding reference.

4. About the method, explain how the studied sample took place. Was it saturation? Explain.

 Regarding saturation, we have introduced in the new manuscript quite a lot of information, both in the results section and in the "limitations of the study" paragraph. In that paragraph we have highlighted that we have not been able to associate the sample size with the saturation of the information and what this entails.

5. On lines 180 to 185 explain: "colleagues and would make it difficult to fulfill their tasks"

 Thank you for your observation. We have rewritten the sentence.

 6. I ask the authors, why didn't they follow COREQ? 

 Thank you very much for this contribution. Consciously, we have not followed COREQ, but we have completed the checklist and realized that we almost did. We have send you  the COREQ checklist. Each issue has been checked and we have included The section and the paragraph where it is mentioned.

In order for the manuscript to reflect the information contained in the COREQ checklist, we have introduced small modifications along the doccument that you can read in the new version.
